# Comparison of scenario reduction approaches for reservoir inflow timeseries generated by a Bayesian Neural Network

Ja-Ho Koo[1,2,3]*, Edo Abraham[1], Andreja Jonoski[2], Dimitri P. Solomatine[1,2,4]

**1** Department of Water Management, Delft University of Technology, CN, Delft, The Netherlands,
**2** Department of Hydroinformatics and Socio-Technical Innovation, IHE Delft, AX, Delft, The Netherlands,
**3** Korea Water Resource Public Corporation, Daejeon, Republic of Korea, **4** Department of river basins hydrology, Water Problems Institute of RAS, Gubkina 3, Moscow, Russia

* koojh78@gmail.com

## Abstract

Dealing with uncertainty in predicted inflows presents a major challenge in optimal reservoir flood control. Scenario-based stochastic control approaches address this by generating multiple inflow time series from probabilistic models, each representing a possible future with associated likelihoods. However, using too many scenarios increases computational complexity, while too few may compromise representativeness. Although the two critical steps of scenario generation and reduction have been extensively explored in other fields, their application to reservoir inflow dynamics remains limited. This study develops and applies a probabilistic data-driven model, specifically, a Bayesian Neural Network (BNN), for scenario generation. While the model exhibits limitations in predicting peak inflows due to data scarcity, it effectively captures temporal dependencies in inflow time series and achieves high short-term accuracy, as measured by the Nash–Sutcliffe Efficiency Coefficient (NSE) and Root Mean Squared Error (RMSE), though performance declines over longer horizons. For scenario reduction, four distance measures widely used in other domains, i.e., the Manhattan, Euclidean, Wasserstein, and energy distances, are evaluated. Experimental results show that the energy distance best preserves the statistical properties of the full scenario set, followed by the Manhattan and Euclidean distances. However, in terms of retaining extreme inflow scenarios, which are critical for flood control, the Manhattan and Euclidean distances outperform others based on a custom index measuring the envelope size of the original scenario set using the $l_1$-norm. In terms of computational efficiency of scenario reduction approaches, the energy distance is the most expensive (quadratic in $m$, the number of reduced scenarios), while the Wasserstein scales linearly. In the examples used, reduced sets are shown to adequately capture extremes when the number of scenarios $m \geq 30$. Considering the trade-off between preserving extremes and computational cost, the Manhattan and Euclidean

**Data availability statement:** All data and code files are freely available from the 4TU.ResearchData repository (https://doi.org/10.4121/e343331b-496f-40ab-83eb-f546df6dffa6).

**Funding:** The author(s) received no specific funding for this work.

**Competing interests:** The authors have declared that no competing interests exist.

distances with $m = 30$ are recommended as a practical choice for reservoir inflow scenario reduction.

## Introduction

Model-based optimal reservoir flood control, typically implemented via model predictive control (MPC) [1], requires predictive regulation of discharge based on forecasts of inflows. Uncertainty in the control operation is therefore mainly due to uncertainty in the prediction of inflows into a reservoir [2], which are driven by meteorological uncertainties. The inflow uncertainty in MPC can be represented as a limited number of feasible future scenarios [3]. Therefore, scenario generation approaches aim to generate discrete time-series that can represent this uncertainty. In the case of stochastic optimization-based MPC approaches, the number of scenarios has a profound impact on the computational complexity; therefore, it is desirable to reduce the number of scenarios while at the same time preserving enough scenarios that 'sufficiently' represent the inflow uncertainty [4,5].

Scenario generation for hydrological variables has traditionally relied on statistical time-series models such as AutoRegressive–Moving Average (ARMA) and AutoRegressive Integrated Moving Average (ARIMA), or on perturbations of historical inflows [6]. These approaches can reproduce general variability but often oversimplify uncertainty, for example, by adding random noise or uniformly perturbing model predictions. Scenario trees have also been used to represent branching future inflows, but they become computationally impractical for reservoir flood control, where multi-step inflow trajectories lead to high dimensionality [7,8].

More recently, data-driven models have been explored in various fields to generate realistic scenarios. Deep learning architectures, such as Convolutional Neural Networks (CNNs) and Long Short-Term Memory networks (LSTMs), can model nonlinear patterns but are usually applied in deterministic settings. Probabilistic methods like Quantile Regression Deep Neural Networks (QRDNNs) capture conditional distributions, yet they struggle to maintain temporal dependence in multi-step forecasts [9,10]. In contrast, probabilistic models such as Gaussian Processes and Bayesian Neural Networks (BNNs) directly estimate predictive parameter distributions, making them well-suited for generating inflow scenarios that preserve both uncertainty and temporal dependence [11,12]. While a few studies have applied BNNs to hydrological prediction, their use for generating multi-step probabilistic inflow scenarios is still uncommon, leaving a methodological gap that this study aims to address.

To clarify, extending deterministic data-driven models to produce probabilistic multi-step scenarios would require additional stochastic mechanisms, e.g., Bayesian layers, ensemble sampling, or generative components, and substantially broaden the methodological scope. Since the objective of this study is to evaluate the applicability of a probabilistic model—rather than compare different deep learning architectures—we focus on BNNs as a representative probabilistic approach.

Even though a large number of scenarios can better represent inflow uncertainty, using too many makes stochastic optimization computationally infeasible. Scenario reduction techniques address this by selecting a smaller subset that still preserves the key properties of the original scenario set. Clustering-based methods, such as k-means and k-median, as well as forward or backward selection methods based on the Wasserstein distance [13], have been widely explored in stochastic optimization. Studies in power and energy systems have shown that k-means can provide robust performance, while Wasserstein-based forward selection often yields lower operational cost [14,15].

Despite these advances, important gaps remain for hydrological applications. Most research has focused on Euclidean and Wasserstein distances [5,16], whereas the Manhattan and energy distances are much less studied, even though they are also widely used distance measures. Moreover, few studies have examined which distance measures best preserve extreme inflow events, an essential requirement for reservoir flood control. Existing evaluations are typically embedded within full stochastic optimization experiments [14,17,18], making it difficult to isolate the performance of the reduction method itself.

Therefore, it is necessary to independently assess how well reduced scenario sets retain both the statistical characteristics and the extreme-event behavior of the original inflow ensemble [19,20]. This is especially relevant in practical flood control, where resilient reservoir operation depends on accurately capturing inflow uncertainty and its extremes.

In this study, the Manhattan, Euclidean, Wasserstein, and energy distances are selected because they are widely used, computationally efficient, and conceptually representative of two major classes of scenario reduction methods, which are vector distance-based and probabilistic distribution distance-based approaches. Their efficiency and transparency make them suitable for repeated online reduction in reservoir flood-control settings, where real-time computation is essential. Although more advanced scenario reduction algorithms exist, such as hierarchical clustering or entropy-regularized optimal transport algorithms, these methods ultimately rely on the same underlying distance measures used in this study. Focusing on the Manhattan, Euclidean, Wasserstein, and energy distances is therefore appropriate, as they constitute the core metrics underlying a wide range of reduction algorithms and provide a meaningful basis for evaluating the applicability of scenario reduction to reservoir inflow ensembles.

To the best of our knowledge, although Bayesian Neural Networks (BNNs) have been applied in some hydrological prediction studies, their use for multi-step probabilistic reservoir inflow scenario generation remains limited, and no prior work has evaluated their suitability within a broader scenario-reduction framework. The novelty of this study does not lie in introducing BNNs themselves, but rather in providing the first systematic integration of (i) probabilistic data-driven inflow scenario generation and (ii) a quantitative comparison of multiple scenario reduction techniques tailored to reservoir flood-control needs. In particular, previous research has not jointly examined these components with explicit attention to extreme-event preservation, which is operationally crucial yet rarely assessed.

This study addresses two aspects that remain underexplored: (i) the statistical representativeness of reduced scenario sets and (ii) their ability to retain extreme inflow trajectories that strongly influence reservoir flood control decisions. The specific contributions are as follows:

1. **Applying and evaluating** a Bayesian Neural Network to generate probabilistic multi-step inflow scenarios and examine their uncertainty propagation and temporal dependence.

2. **Conducting a systematic comparison** of four widely used scenario reduction distance measures—the Manhattan, Euclidean, Wasserstein, and energy distances—using metrics designed to assess both statistical fidelity and extreme-event preservation.

3. **Providing practical guidance** on selecting an appropriate number of reduced scenarios and choosing distance measures suited for real-time reservoir flood control, considering the balance between computational efficiency, statistical quality, and the retention of extreme scenarios.

This paper is organized into four sections. The Method section introduces the methodologies for scenario generation and reduction, focusing on Monte-Carlo dropout BNN and four distance measures. The Results and Discussion section details the case study area and presents the experimental results, including evaluating a reduced scenario set based on the $l_1$-norm to emphasize the inclusion of extreme scenarios. Finally, the conclusions summarize the key findings and implications of this study.

## Method

### Scenario generation by Monte-Carlo dropout BNN

From the stochastic programming perspective, the scenario approximates a continuous probability distribution of stochastic variables with a discrete distribution [21]. Hydrological factors such as reservoir inflow and water level are continuously valued, and temporal dependence is critical. Therefore, the hydrological multi-period scenarios should be generated using a model in which temporal dependence can be captured. In this context, data-driven models such as Gaussian processes or BNN that can derive probabilistic hydrological factors are suitable for generating hydrological scenarios [22]. However, compared to conventional Gaussian processes, which assume that the probability of uncertainty follows a Gaussian distribution, BNN can easily utilize arbitrary distributions and is known to be efficient for learning and evaluating large-scale data [23].

A BNN is an artificial neural network trained to derive probabilistic outputs based on Bayesian inference [11]. In BNN, each parameter $\theta$ in Eq 1a, which is traditionally treated as deterministic in Deep Neural Networks (DNNs), follows a specific probability distribution, as follows:

$$\tilde{\mathbf{y}} = F(\mathbf{x}; \theta),$$

(1a)

$$\theta \sim p(\theta),$$

(1b)

where $p(\theta)$ is the probability distribution of each element of $\theta$. BNN learns a probability distribution $p$ of $\theta$ during the training process. However, estimating the posterior probability of parameters is computationally intensive and time-consuming [12]. As a result, [12] proposed a Monte-Carlo (MC) dropout method. By randomly deactivating some nodes (dropout) and repeating random dropouts (Monte-Carlo simulation), multiple estimated outputs are generated. These multiple outputs are uncertain scenarios. The probability distribution of the outputs can be estimated. Thus, MC dropout BNN can be considered a BNN with a Bernoulli distribution for every $\theta$ [12,24]. Given the straightforward implementation of a dropout technique, this method has low complexity and enables fast approximation [25,26].

### Scenario reduction

Let an original scenario set be $Y(\mathcal{Y}, \mathbf{w})$, where $\mathcal{Y} = (\mathbf{y}_1, ..., \mathbf{y}_n)$ with $\mathbf{y}_i \in \mathbb{R}^k$ with corresponding probabilities $\mathbf{w} = (w_1, ..., w_n)$, where $n$ is the number of original scenarios, and a reduced scenario set be $X(\mathcal{X}, \mathbf{v})$, where $\mathcal{X} = (\mathbf{x}_1, ..., \mathbf{x}_m)$ with $\mathbf{x}_i \in \mathbb{R}^k$ with corresponding probabilities $\mathbf{v} = (v_1, ..., v_m)$, where $m$ is the number of reduced scenarios and typically $m \ll n$. Then, scenario reduction is a technique that finds $X(\mathcal{X}, \mathbf{v})$ that well represents $Y(\mathcal{Y}, \mathbf{w})$, where $\mathcal{X} \subset \mathcal{Y}$ for discrete scenario reduction or $\mathcal{X} \not\subset \mathcal{Y}$ for continuous reduction [5,16]. Here, 'representing' is measured via closeness in some distance metric, and the reduction approach aims to minimize the distance between sets. Therefore, the choice of an appropriate measure for the distance between scenarios is crucial. In general, the distance between scenarios ($\mathbf{y} \in \mathbb{R}^N$ and $\mathbf{x} \in \mathbb{R}^N$) is calculated by the Manhattan distance ($l_1$ metric) and the Euclidean distance ($l_2$ metric).

Because of their superior ability to capture differences in the shape of distributions compared to other alternatives (here between $\mathcal{Y}$ and $\mathcal{X}$), both the Wasserstein distance [16] and energy distance [27] are well-established metrics for

comparing probability distributions. This is also demonstrated through their widespread use in machine learning and stochastic optimisation research [28].

The two classical distance metrics are defined using the $l_1$-norm and $l_2$-norm between corresponding time series vectors of each scenario, respectively, as follows:

$$d_l(\mathbf{x}, \mathbf{y}) = \|\mathbf{x} - \mathbf{y}\|_l = \left(\sum_i |x_i - y_i|^l\right)^{1/l},$$

(2)

where $d_l$ denotes the Manhattan distance when $l = 1$ and the Euclidean distance when $l = 2$ between scenarios, $\mathbf{x}$ and $\mathbf{y}$, in which $x_i$ and $y_i$ are the $i^{th}$ elements of each scenario.

Scenario reduction using the Manhattan and Euclidean distances utilizes the widely used clustering algorithm. The original scenarios are divided into $m$ clusters, and the cluster centroids serve as reduced scenarios. K-median is a representative clustering algorithm using the Manhattan distance [29], and k-means is a popular clustering algorithm using the Euclidean distance [30]. K-median is more effective for data with many outliers and asymmetric distributions due to the $l_1$-norm. Nevertheless, computational complexity is a disadvantage compared to k-means clustering.

An additional distinction is that the centroid in a cluster of k-median is the element-wise median of scenarios in a cluster, whereas the k-means' centroid is the element-wise mean. Every centroid is a reduced scenario. Therefore, a reduced scenario set contains a new set of time series chosen continuously in $\mathbb{R}^k$ (i.e., $\mathcal{X} \not\subset \mathcal{Y}$), and so the method is also called continuous scenario reduction. The continuous scenario reduction offers flexibility to find reduced sets from infinitely many options from a continuous set. However, it also has issues concerning the validity of the new scenarios [16]. For example, in hydrological scenarios, preserving the temporal dependence within each scenario is essential. Thus, selecting the element-wise median or mean of each scenario subset in clustering may weaken the preservation of the temporal dependence that was present in the original scenarios. Therefore, we modify the algorithms to select a reduction scenario from the original scenarios by identifying the one closest to each cluster's centroid (median or mean), measured by the corresponding distance measure, as follows:

$$\mathbf{x}_i = \arg\min_{\mathbf{y}_j \in C_i} \|\mathbf{y}_j - \boldsymbol{\mu}_i\|_l,$$

(3)

where $\mathbf{x}_i$ is a reduced scenario of cluster $i$, $\boldsymbol{\mu}_i$ is the centroid of cluster $C_i$ (element-wise median for the Manhattan distance when $l = 1$, and element-wise mean for the Euclidean distance when $l = 2$). As a result, this approach differs from conventional k-means and/or k-median clustering in that the calculated cluster centroids are not directly used as the reduced scenarios.

The probability of a centroid of cluster $C_i$ by k-median and k-means is defined as follows:

$$\Pr(\mathbf{x}_i) = \frac{|C_i|}{n},$$

(4)

where $|C_i|$ is the number of scenarios in $C_i$ and $n$ is the total number of original scenarios.

Conversely, the Wasserstein distance and energy distance can be used to calculate the distance between two probability distributions. The Wasserstein distance can be expressed as the minimum (transport) cost required to make two probability distributions identical, as follows [31]:

$$d_{W,p}(X, Y) = \left(\inf_{\gamma \in \Gamma(X,Y)} \int_{\mathbb{R}^k \times \mathbb{R}^k} \|x - y\|^p \, d\gamma(x, y)\right)^{1/p},$$

(5)

where $X$ and $Y$ represent the probability distributions of different scenario sets. $d_{W,p}$ is the $p$-Wasserstein distance, and $\Gamma(X, Y)$ is the joint probability distribution of $X$ and $Y$.

The energy distance is defined as Eq 6 [5,27].

$$d_{E,p}(X, Y) = 2\mathbb{E}\|X - Y\|_2^p - \mathbb{E}\|X - X'\|_2^p - \mathbb{E}\|Y - Y'\|_2^p, \tag{6}$$

where $X'$ is the independent and identically distributed (i.i.d.) copy of scenario distribution $X$. The energy distance effectively captures comprehensive distributional differences by considering both the inter-distributional and intra-distributional distances.

For these two distance measures, scenario reduction involves finding a subset $X$ of the original scenario set $Y$ that minimizes $d_W$ or $d_E$, as follows:

$$X = \arg\min_{X \subset Y} d(X, Y), \quad \text{where } |X| = m, \tag{7}$$

where $d(X, Y)$ represents either $d_W$ in Eq 5 or $d_E$ in Eq 6, and $m$ is the number of reduced scenarios.

The exact computation of the Wasserstein distance is computationally complex. However, Eq 7 can be simplified to a (mixed-integer) linear program [16,32]. The work in [33] suggested the optimal redistribution rule, where the probability of a reduced scenario equals the sum of the original probabilities of the unselected scenarios proximate to this reduced scenario. This allows us to calculate reduced scenarios and their probabilities without explicitly solving a mixed-integer linear program [5,33]. However, the energy distance-based scenario reduction in Eq 7 is computationally disadvantageous compared to the Wasserstein distance, as it requires solving a quadratic program with linear constraints [5].

It is worthwhile to clarify that different algorithms are utilized for different distance measures. This is due to the different properties of each distance measure. The Manhattan and Euclidean distances are the distances between vectors, i.e., each scenario. However, the Wasserstein and energy distances can be measured between probability distributions, i.e., a distribution of original scenarios and a distribution of reduced scenarios. This difference leads to the utilization of different algorithms. The distances between two scenarios to sort scenarios, i.e., clustering, and pick one representative scenario in each cluster can be utilized. In contrast, the similarity between the original scenario set and the reduced set can be compared by the Wasserstein or energy distance [5,34]. Scenarios can be added one by one to the initially empty reduced set, each selected to minimize the distance between the two distributions (forward selection). On the other hand, scenarios can be removed from the reduced set, which initially contains all original scenarios (backward selection). Alternatively, the reduced scenario set can be found directly by calculating all possible combinations of reduced scenarios and selecting the one with the smallest distance from the original scenarios.

When the forward or backward selection method is applied using the Manhattan or Euclidean distance, it is clear that only the scenario closest to the mean is selected at each step. Furthermore, it is not possible to apply any clustering algorithms using the Wasserstein or energy distance. Therefore, it is evident that we only have limited feasible algorithms once a distance measure is selected. In addition, given that applying the same algorithm for all distance measures is not possible, it would be better to apply the 'best' or 'widely-used' algorithm for each distance measure to ensure a fair comparison. This is why we utilize the k-means for the Euclidean, k-median for the Manhattan, and forward selection for the Wasserstein and energy distance.

The method presented in this section is illustrated in Fig 1. Scenario generation is performed using MC dropout BNN, and the resulting (very large number of) scenarios are reduced using four distance measures. These reduced scenarios are then evaluated based on criteria that best replicate the original generated scenarios. The details of this process are provided in the following sections.

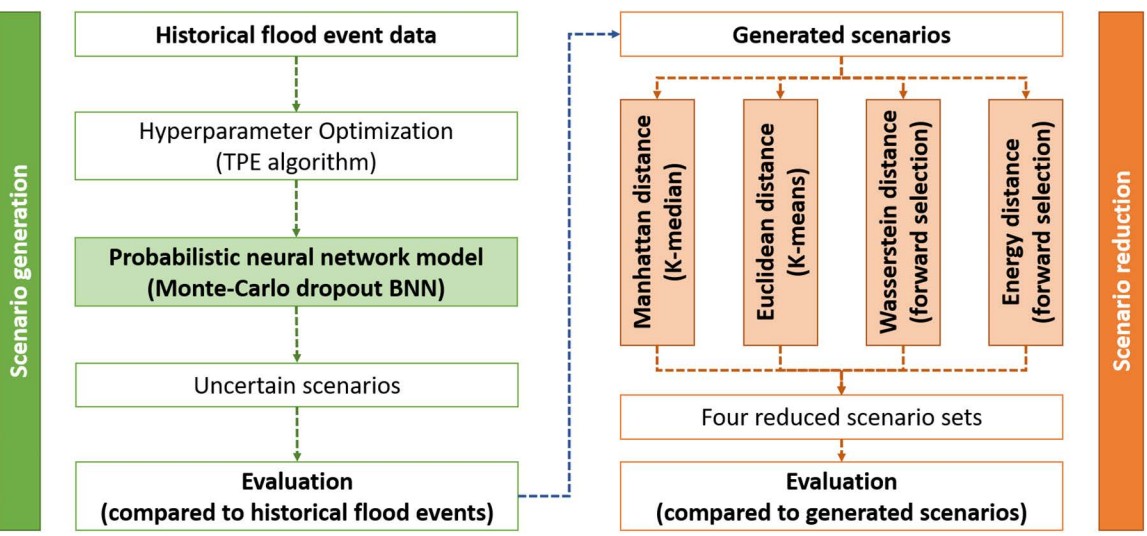

**Fig 1. Flowchart of the study framework, comprising scenario generation and scenario reduction.** The methods are applied to reservoir flood inflow time series to assess their practical applicability.

## Results and discussion

### Flood inflow at the Daecheong reservoir in Korea: Case study

The Daecheong Reservoir, situated in the central region of the Geum River in Korea, has maintained an extensive observational database (Fig 2). Quality-controlled hourly data are made available by the management authority, the Korea Water Resources Public Corporation (K-water). These officially published data have no missing values or outliers since K-water has cleaned the data and checked its precision.

Despite the availability of sufficient data for the Daecheong reservoir, we used hourly data from 2011 to 2020 due to limited observations from upstream water level stations. Since prediction models struggle to accurately identify inflection points using only observed inflow data, upstream water levels are included as additional input features to help the model implicitly learn flow trend changes. There are a total of 22 water level stations in the upper stream, but only observed data from 16 water level stations are available for the study period, as presented in Fig 2. Other stations were either installed after 2011 or have missing data, even in their officially published records. Unlike reservoir data, which contained no missing data or outliers, official water level data can contain missing data and outliers. This is reasonable because data quality regulations for reservoirs do not allow missing data, but for water level stations, data with missing values or outliers are allowed to be published even when they represent actual observations in Korea. Reservoir data are more strictly managed due to both their importance and the straightforward data production process. Further information is available in [35]. In this study, instead of cleaning unofficial data ourselves, we only utilize data from stations without missing values or outliers. All data are obtained from the public data portal (www.data.go.kr) operated by the Korean government and K-water (www.kwater.or.kr).

Since we aim to generate and reduce scenarios for flood events, it would be appropriate to use data exclusively from flood events. However, with only nine flood events available, the data are insufficient to train a BNN model. Accordingly, all hourly data of reservoir inflow and upstream water levels from September 2012–2020 are used to train and validate the BNN model. Despite the limited number of observed extremes, the probabilistic scenario ensemble can still cover peak-like inflow behavior through stochastic variability. For the validation, 20% of the data are randomly selected. Note that, because the model generates a complete multi-step inflow scenario at each time step using only information

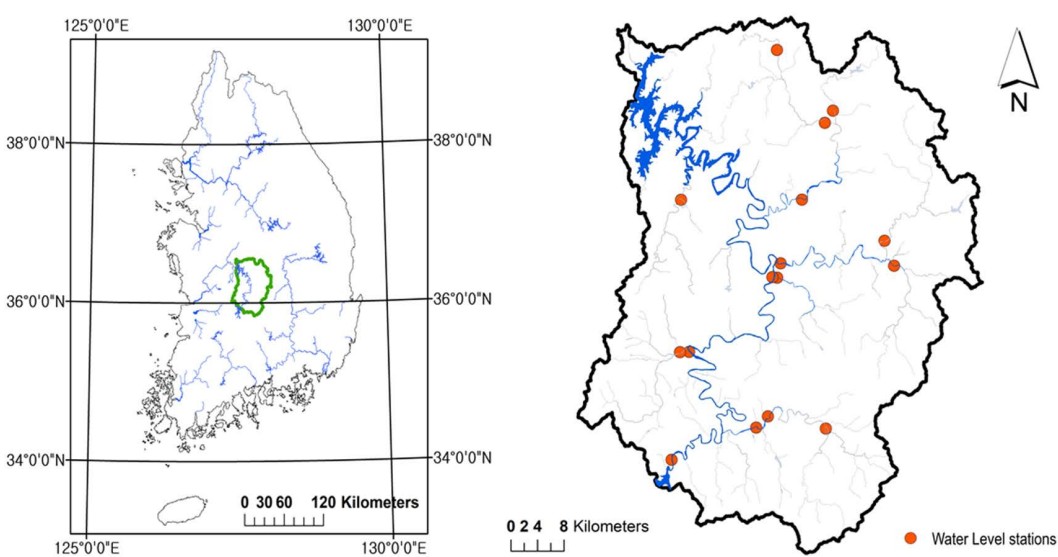

**Fig 2. Location of the Daecheong(DC) reservoir and upstream water level stations.** The left map shows the DC basin area. The right figure shows the location of the Daecheong Reservoir and water level stations in the upstream area. The basin and river shape files of Korea are from the public data portal in South Korea (www.data.go.kr) and VWorld (https://www.vworld.kr; access may be restricted outside South Korea due to national security policies), respectively. They were originally generated by the National Geographic Information Institute (https://www.ngii.go.kr) and the Han River Flood Control Office(https://www.hrfco.go.kr/).

available at each time step, the temporal dependency that must be preserved is internal to each scenario vector rather than across different timestamps in the dataset. Therefore, given that only a small number of flood events are available, random sampling for the validation does not induce information leakage in this setting, even though chronological validation is generally recommended in time-series prediction. The model testing, however, is conducted on two specific flood events: a flood event in August 2011 (Event 1: 2011-08-07–2011-08-18, for 259 hours) and another in August 2012 (Event 2: 2012-08-12–2012-08-21, for 221 hours). The rationale for this chronological inversion—training on data from September 2012–2020 and testing on preceding events—stems from the distribution of major flood events within the available dataset. The most significant flood events occurred at the beginning and the middle of the overall observation period. To ensure that both the training and testing datasets included substantial flood dynamics, we deliberately distributed these extreme events across the sets. This specific partitioning allows the model to learn from severe floods during the 2012–2020 period while being rigorously evaluated on distinct, unseen extreme events from 2011 and early 2012, thereby preventing data leakage and effectively assessing the model's physical generalization capabilities (i.e., hindcasting).

## Scenario generation by BNN

An MC dropout BNN model for scenario generation has $N$ output nodes corresponding to the prediction horizon $N$. Reservoir inflow, spatial mean rainfall, and the upstream water levels from 16 stations during an autoregressive horizon $B$ constitute the input features, so the number of input nodes is $B \times (1 + 1 + 16)$. While a smaller $N$ can enhance the BNN model performance [36], it compromises the effectiveness of reservoir flood control optimization in a receding horizon framework compared to a large $N$ [1]. Therefore, in this study, $N$ is set to 12.

The performance of DNN models depends heavily on hyperparameters [37,38]. Hyperparameters of the probabilistic inflow model are optimized using the Tree-structured Parzen Estimator (TPE) algorithm [38,39], a widely used Bayesian

optimization approach. TPE models the performance of past hyperparameter trials and proposes new candidates by favoring configurations expected to improve model accuracy. The Root Mean Square Error (RMSE) of the validation set is used as the objective function. All optimization is implemented using the Optuna library [40].

To reduce the number of hyperparameters to be optimized, the early-stopping technique [41] is utilized, terminating the training process when validation loss shows no improvement over a specified number of epochs. Additionally, the number of past values $B$ is also optimized at the same time. The optimal hyperparameters are presented in Table 1. The optimal autoregressive horizon is 24, resulting in a total of 24 × 18 input nodes. The optimal dropout rate is 10%, with 512 nodes in the hidden layer, a learning rate of 0.0005, and a batch size of 64. The selected activation function is the Rectified Linear Unit (ReLU). The architecture of the BNN is illustrated in Fig 3.

The MC dropout BNN model is evaluated using the Root Mean Square Error (RMSE) and the Nash-Sutcliffe model Efficiency coefficient (NSE), which are widely used to evaluate the reliability of a hydrological prediction model, as demonstrated in Fig 4 [42–45]. In reservoir flood control, reducing significant deviations, such as peak inflows, is more important than minimizing minor discrepancies. Therefore, RMSE appears to be a suitable metric, as it places greater emphasis on larger errors. For test events, the RMSE is 112.5 $m^3/s$, given that the mean inflow over the entire period is 106.2 $m^3/s$ and the peak inflow is 3557.0 $m^3/s$. In addition, the NSE is 0.736, considering that 0.65 or higher NSE is generally judged as 'good' and 'very good' for over 0.75 in the hydrological prediction model [46]. Overall, the RMSE and NSE indicate the reliable performance of this model.

In particular, as shown in Fig 4, the BNN model shows a degradation of hydrological prediction performance with increasing prediction horizon, i.e., increasing the RMSE and decreasing the NSE. For example, the RMSE increases from 83.3 m³/s (NSE: 0.885) for 1-hour predictions to 149.4 m³/s (NSE: 0.432) for 12-hour predictions.

However, the prediction performance also degrades for the peak inflow, as illustrated in Fig 5. This issue is also evident in several studies on flood inflow prediction, as considerable errors can be observed in their results (e.g., [47–49]). In this figure, the predicted peak inflow of the scenarios' mean for Event 1 is 3364.3 m³/s, which is close to the observed peak of 3557.0 m³/s. However, for Event 2, the predicted peak inflow of the scenarios' mean is 2354.4 m³/s, significantly lower than the real one of 3093.0 m³/s. This limitation is particularly significant considering the importance of peak inflow during flood events for reservoir flood control.

This degradation is mainly attributed to the limitation of training data, i.e., the lack of sufficient data with high inflow. While we utilize the data from the entire period for model training, the majority of cases have peaks that are much smaller than shown for Event 1 and Event 2 in Fig 5. Typical inflow patterns feature a modest rise in upstream water level and inflow for a brief period, followed by a gradual decline. Training the BNN model exclusively on considerable flood events, such as our test events, could enhance prediction accuracy for those extremes. However, due to the small sample size, this would substantially reduce overall model performance and compromise its practical applicability.

Despite the limitations in peak inflow prediction, a key advantage of probabilistic approaches is their ability to capture a wide range of uncertainty. For example, the maximum inflows among all scenarios for Events 1 and 2 are 4393.9 m³/s and 3065.2 m³/s, respectively. This indicates that, although the predicted peak inflow (i.e., the mean of the scenarios)—which would be the output of a typical deterministic model—is substantially lower than the observed peak, the actual inflow values fall within or near the upper bound of the predicted range. This is illustrated in Fig 6 and highlights the importance of retaining extreme scenarios, even when they have low likelihoods, to effectively capture peak inflows during flood events.

**Table 1. Hyperparameter optimisation result.**

| B | dropout rate | nodes | hidden layers | learning rate | batch | activation |
|---|---|---|---|---|---|---|
| 24 | 0.1 | 512 | 3 | 0.0005 | 64 | ReLU |

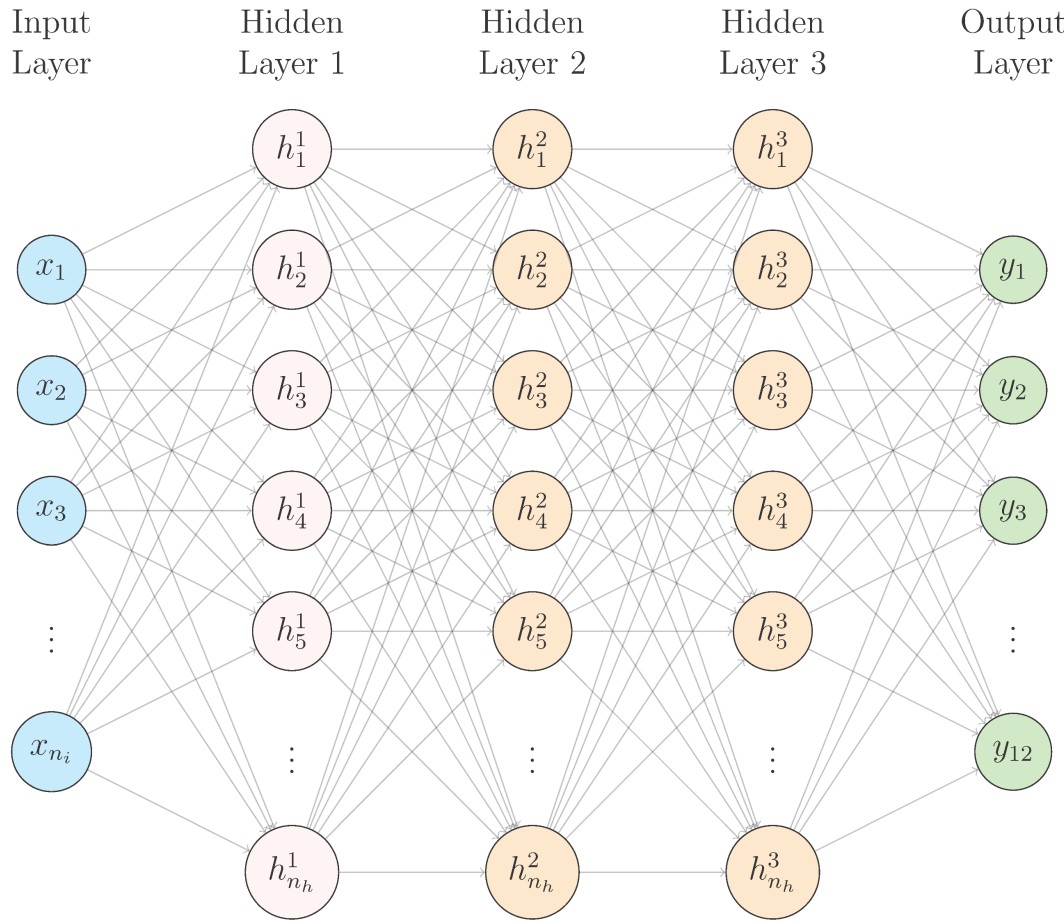

**Fig 3. Architecture of BNN with three hidden layers.** The network consists of $B \times 18$ input nodes ($n_i = B \times 18$), where $B$ is the autoregressive horizon, including past inflow, rainfall and upstream water levels, three hidden layers with 512 nodes each ($n_h = 512$), and 12 output nodes. Connections between each layer are dropped randomly at a dropout rate of 0.1.

Note that this prediction range represents the full envelope defined by the maximum and minimum values of all generated scenarios at each time step, rather than a statistical confidence interval.

In addition to deterministic metrics and graphical analysis, we evaluated the probabilistic performance of the BNN model using the Continuous Ranked Probability Score (CRPS), Prediction Interval Coverage Probability (PICP), and Quantile Loss, as summarized in Table 2. The CRPS and Quantile Loss consistently increase as the prediction horizon extends, correctly reflecting the natural accumulation of uncertainty over time. The PICP for the 95% confidence interval averages around 0.39, which is lower than the nominal level. This indicates that the model is under-dispersed, likely due to the dominance of low-flow data in the training set, which biases the model toward narrower uncertainty bounds.

However, the operational reliability improves significantly when considering the full scenario range. Most crucially, for the total inflow volume ($l_1$-norm), which is a critical factor for reservoir flood control under short-term forecasts, the Min-Max PICP reaches approximately 0.85. This finding highlights again that the extreme scenarios (upper/lower bounds) successfully capture the operational risk space and the importance of preserving the envelope size in scenario reduction.

Currently, our model utilizes standard MSE loss for training and RMSE for validation, which implicitly assumes homoscedasticity (constant observation noise). This assumption limits the model's ability to capture data-dependent

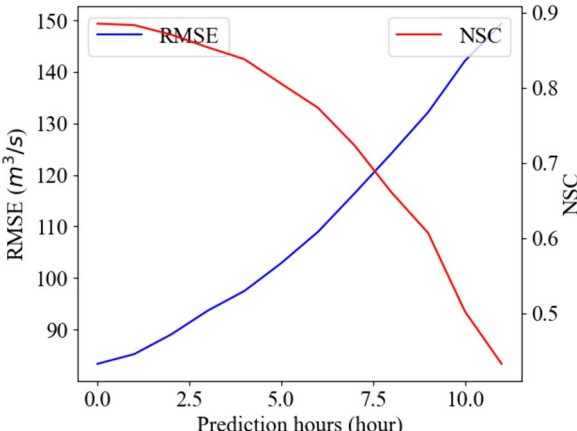

**Fig 4. BNN model performance with prediction length in RMSE and NSE metrics.**

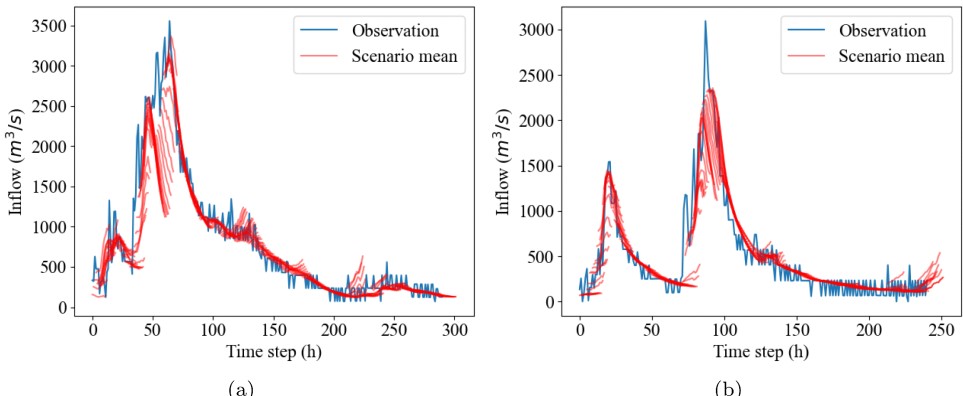

**Fig 5. Real and BNN prediction hydrographs for test events:** (a) Event 1, (b) Event 2.The red lines indicate the scenario means of predictions for the next 12 hours, updated every hour.

aleatoric uncertainty, leading to under-dispersed prediction intervals, particularly during extreme events where variance typically increases. To fundamentally improve probabilistic performance, loss functions that explicitly model both aleatoric and epistemic uncertainties, such as the Gaussian Negative Log-Likelihood (GNLL) [50], are required. However, simultaneously estimating both the mean and data-dependent variance often increases optimization complexity and requires a substantial amount of data to ensure convergence stability. Given the limited number of flood events in our dataset, we prioritized model robustness by employing the standard MSE loss to establish a stable baseline.

## Scenario reduction

Scenario reduction by vector distance-based approaches using clustering algorithms is implemented using the Python library, scikit-learn [51]. For scenario reduction based on the Wasserstein distance and energy distance, we follow the method of [5]. As discussed, the final reduced scenarios $\mathbf{x} \in \mathcal{X}$, which are closest to centroids based on the distance measures, are selected from the original scenarios, to ensure $\mathcal{X} \subset \mathcal{Y}$. This approach differs from conventional

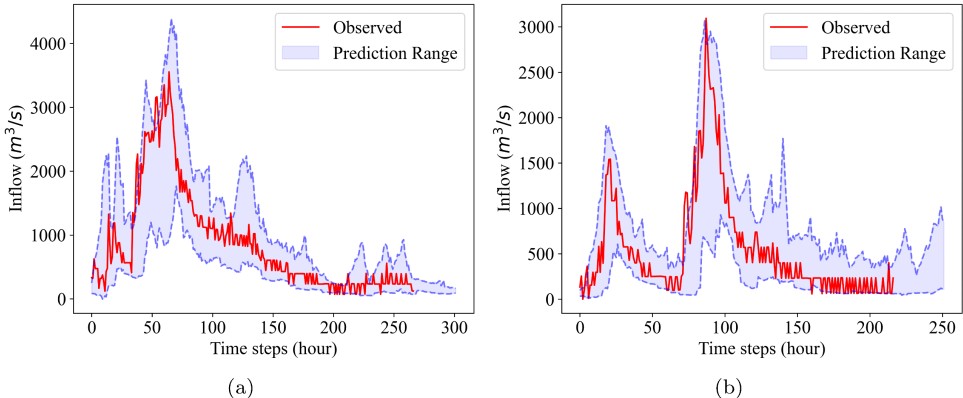

**Fig 6. Real inflows and prediction ranges of BNN models for test events:** (a) Event 1 (b) Event 2.Prediction ranges are defined by the maximum and minimum values across predicted scenarios for each prediction time, generated every hour for the next 12 hours.

**Table 2. Probabilistic performance metrics averaged over flood events. '100%' denotes metrics calculated using the minimum and maximum of the generated scenarios.**

| Time (h) | PICP(95%) | PICP(100%) | CRPS($m^3/s$) | Q-Loss(95%) |
|---|---|---|---|---|
| 1 | 0.43 | 0.64 | 102.6 | 46.1 |
| 6 | 0.39 | 0.65 | 138.8 | 72.6 |
| 12 | 0.34 | 0.61 | 216.9 | 134.1 |
| Average | **0.39** | **0.64** | 150.3 | 78.4 |
| Total Volume($I_1$) | 0.54 | **0.85** | 1282.7 | 757.8 |

clustering-based scenario reduction because it does not use the element-wise median (k-median) or mean (k-means) of each scenario cluster.

The original scenario set $Y$ consists of 1000 scenarios to cover possible uncertain conditions, each represented by a 12-dimensional vector. This number was determined empirically based on preliminary tests; it is large enough to capture the predictive uncertainty and stabilize the empirical distribution, yet small enough to keep the computational cost of the subsequent scenario reduction algorithms manageable. These scenarios are generated at each time step. Considering the high dimensionality of each scenario and the substantial number of original scenarios, a simple 1-step forward selection is applied for scenario reduction with the Wasserstein distance and the energy distance [5,34]. In addition, the parameter $p$ in Eq 5 and Eq 6 is set to one for both distance measures [5].

Testing the scenario reduction for every time step of Event 1 and Event 2 is computationally intensive, particularly due to the computational complexity of the energy distance. Therefore, we examine three representative points from each event (case 1&4: increasing, case 2&5: decreasing, and case 3&6: stable), as shown in Fig 7.

When the number of reduced scenarios is $m = 10$, the scenario reduction results for each case are shown in Fig 8. In the figure, the solid grey lines refer to the original scenarios, the solid blue lines illustrate the reduced scenarios, and the thickness of the solid blue line represents the probability of each reduced scenario.

The water level and outflow from the reservoir during a flood event significantly impact the upstream and downstream flood conditions. Moreover, extreme events can have a considerable impact on dam safety. Therefore, it seems that a critical criterion for flood scenario reduction methods is whether extreme scenarios are included. From this perspective, the Manhattan and Euclidean distances demonstrate superior performance, followed by the Wasserstein distance and the

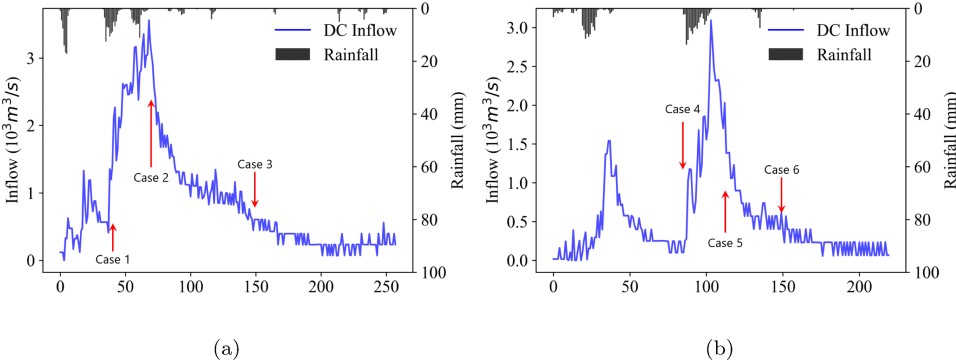

**Fig 7. Scenario reduction test cases in two events:** (a) Event 1 (b) Event 2.

energy distance, as illustrated in Fig 8. Specifically, when using the energy distance and the Wasserstein distance, sparse extreme scenarios tend to be excluded.

Let's examine this in detail. To evaluate this, estimating a scenario envelope and comparing its size with the envelope size of the original scenarios is necessary. Although it is difficult to estimate the exact envelope due to the possibility of the existence of intersecting scenarios, we approximate the size of the envelope using $l_1$-norm as follows:

$$\left|\text{env}(Q)\right| = \max_{q \in Q} \|q\|_l - \min_{q \in Q} \|q\|_l,$$

(8)

where $\left|\text{env}(Q)\right|$ represents the size of the envelope of a scenario set $Q$, which can be an original or reduced set. Here, we set $l$ to one.

The upper envelope encapsulates the maximum possible inflow coming in, and the lower envelope encapsulates the minimum inflow. These represent the flood risk space as a function of reservoir inflow and are of critical importance. Moreover, in the context of short-term flood scenarios, predicting the total inflow volume could be more important than precisely estimating each element of the inflow time series. The $l_1$-norm of each scenario represents the total inflow volume, due to the fact that all inflow elements are positive values. While conventional distributional metrics prioritize the overall statistical fit and may discard rare extreme events as outliers, reservoir flood control requires explicit awareness of the worst-case boundaries. Therefore, the envelope size defined via the $l_1$-norm serves as an important additional metric to measure how well the reduced set preserves the operational safety margins. Therefore, the use of the $l_1$-norm is reasonable. This envelope volume, i.e., the span of flood risk in the scenario set, can be used as a metric of goodness for assessing the reduced scenario set.

We analyze $\left|\text{env}(Q)\right|$, which is the difference between scenarios with maximum and minimum sums, for five different numbers of reduced scenarios ($m = 10, 20, 30, 40,$ and $50$), as illustrated in Fig 9. As described in Fig 8, the Manhattan and Euclidean distances preserve extreme scenarios even with small $m$ values. For example, in Case 3, the envelope volumes obtained using the Manhattan and Euclidean distances are approximately $2.5 \times 10^3 m^3/s$, which are significantly larger than those obtained using the Wasserstein and energy distances, both of which are less than $1.5 \times 10^3 m^3/s$, when $m = 10$. The energy distance tends to focus more on the scenarios that frequently appear. All distance measures include more extreme scenarios as the number of reduced scenarios $m$ increases. The result suggests that reduced scenario sets adequately capture extreme scenarios when $m \geq 30$. It is worth noting that this metric, $\left|\text{env}(Q)\right|$, does not inherently favor the Manhattan or Euclidean distances. Although it employs the $l_1$-norm to estimate total inflow volume, this use is

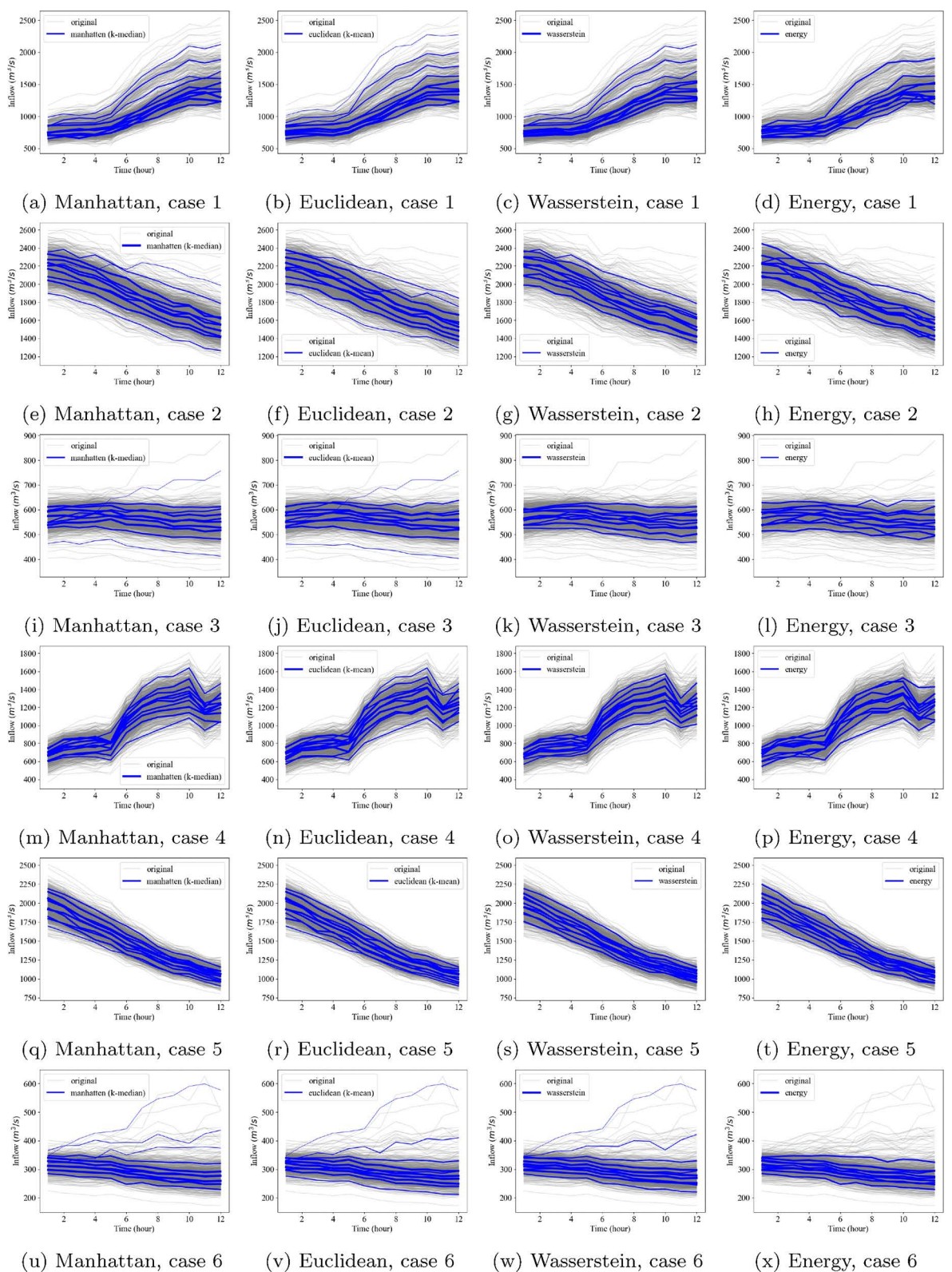

(a) Manhattan, case 1  (b) Euclidean, case 1  (c) Wasserstein, case 1  (d) Energy, case 1

(e) Manhattan, case 2  (f) Euclidean, case 2  (g) Wasserstein, case 2  (h) Energy, case 2

(i) Manhattan, case 3  (j) Euclidean, case 3  (k) Wasserstein, case 3  (l) Energy, case 3

(m) Manhattan, case 4  (n) Euclidean, case 4  (o) Wasserstein, case 4  (p) Energy, case 4

(q) Manhattan, case 5  (r) Euclidean, case 5  (s) Wasserstein, case 5  (t) Energy, case 5

(u) Manhattan, case 6  (v) Euclidean, case 6  (w) Wasserstein, case 6  (x) Energy, case 6

**Fig 8. Scenario reduction results when *m* = 10.**

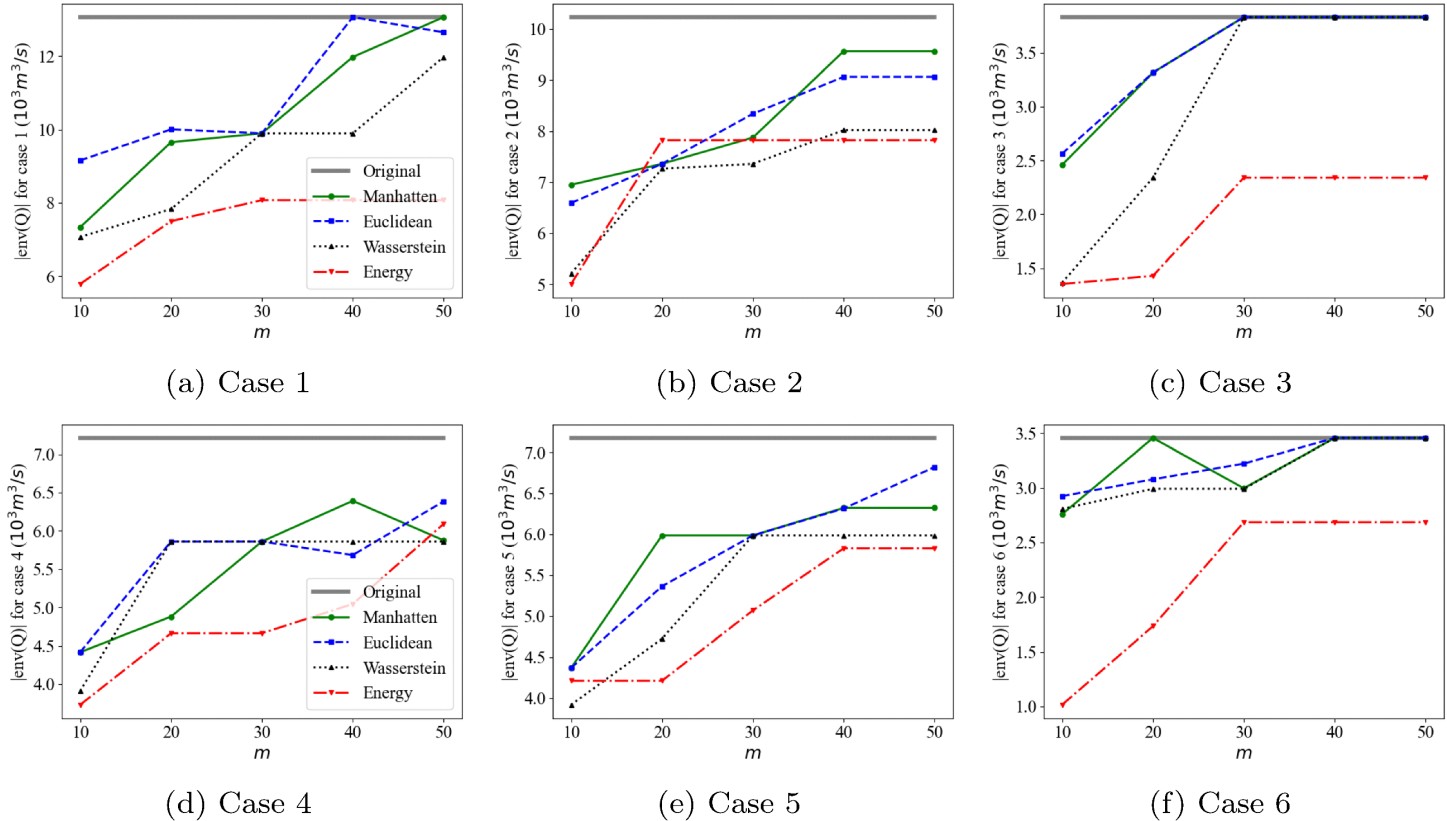

**Fig 9. The size of the envelopes of the original and reduced scenario sets.**

conceptually distinct from how the Manhattan and Euclidean distances are applied in scenario reduction, and so the performance differences are not a priori expected when comparing all four distance measures.

To understand why vector distance-based measures (Manhattan and Euclidean) better preserve extreme scenarios compared to probabilistic measures, one must examine the respective objective functions. Vector-distance approaches, implemented via clustering algorithms, aim to minimize the spatial dispersion of data points. The objective functions for k-means (Euclidean) and k-median (Manhattan) are defined as $J_{means} = \sum_{j=1}^{m} \sum_{y_i \in C_j} \|y_i - \mu_j\|_2^2$ and $J_{median} = \sum_{j=1}^{m} \sum_{y_i \in C_j} \|y_i - \mu_j\|_1$, where $m$ is the number of reduced scenarios, $C_j$ is the $j$-th cluster, and $\mu_j$ is the centroid. Although the $l_1$-norm of k-median is generally less sensitive to outliers than the squared $l_2$-norm, both algorithms are fundamentally *space-partitioning* methods. An extreme flood scenario is geometrically distant from the central cluster. To minimize the total aggregated distance via both norms, the algorithm is likely to choose a representative centroid near these distant points; otherwise, the accumulated spatial error would remain significantly high.

In contrast, probabilistic measures like the Wasserstein distance incorporate the *probability mass* of each scenario into the cost function. Conceptually, the cost contribution of a scenario $x_i$ relates to the transport cost weighted by its probability, as $Cost \approx \sum \|y_i - x_j\| \cdot w_i$, where $w_i$ is the probability of the original scenario $y_i$. Since extreme flood events are rare, their associated probability $w_i$ is very low (e.g., $p \approx 0$). Consequently, even if the geometric distance $\|y_i - x_j\|$ is large, its contribution to the total probabilistic cost is negligible. Therefore, probabilistic algorithms tend to "sacrifice" the preservation of these low-probability extremes to better fit the high-density regions near the distribution's center (the mean).

Other suitable and widely used metrics for comparing probability distributions of time-series data [5] and specifically hydrological data [52] are the mean ($\mu$) and the standard deviation ($\sigma$). This is also because the uncertainty of inflow time series, e.g., residuals for model uncertainty, has been modelled using a logistic distribution or a normal distribution [53,54], whose statistics can therefore be completely described by the mean and standard deviation. The mean and standard deviation of the original and reduced scenarios are compared, as shown in Fig 10. Regarding scenario means, the energy distance demonstrates the closest alignment with the original scenarios and the lowest Mean Absolute Errors (MAE) of means in Fig 10b, which is 2.1 $m^3/s$ when $m = 10$ and decreases to 0.4 $m^3/s$ when $m \geq 40$. The Euclidean distance exhibits the largest differences, together with the Manhattan distance. Notably, the maximum difference in means is 3.5 $m^3/s$ when

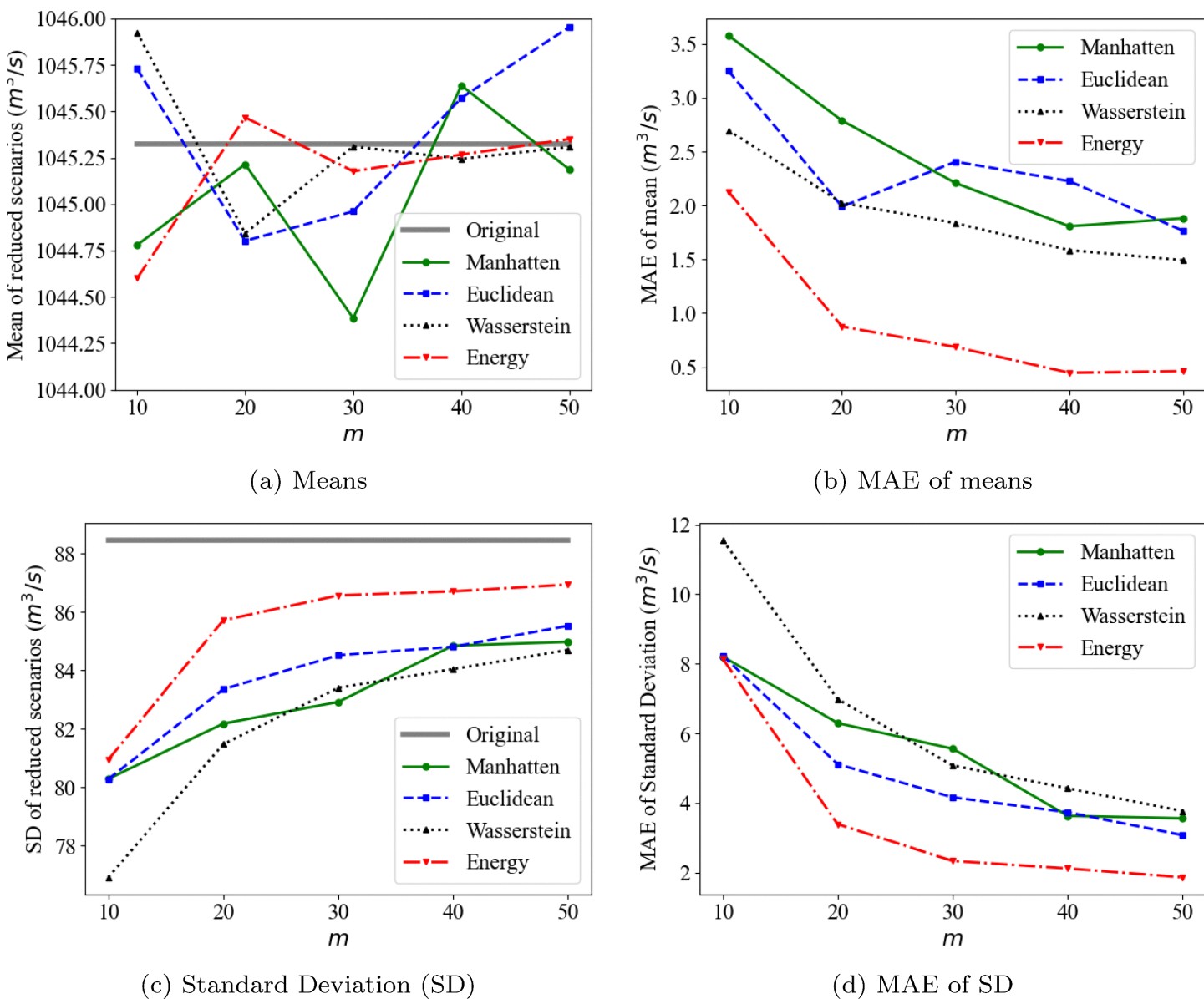

(a) Means

(b) MAE of means

(c) Standard Deviation (SD)

(d) MAE of SD

**Fig 10. Means and Standard deviations of original and reduced sets.**

$m = 10$ for the Manhattan distance, representing only 0.3% of the mean of 1045.3 $m^3/s$ in the original scenarios, as shown in Fig 10a.

In terms of standard deviation, the energy and Euclidean distances demonstrate the minimum differences, as illustrated in Fig 10d. Notably, unlike the means of reduced scenarios, which show no clear relationship with $m$ in Fig 10a, the standard deviations across all distance measures gradually converge to the original scenarios' $\sigma$ as $m$ increases, as shown in Fig 10c. For instance, the MAE of the standard deviations of the Wasserstein distance decreases from 11.6 $m^3/s$ to 3.8 $m^3/s$.

Regarding time-series scenarios, temporal correlation is one of the key characteristics, along with mean and variance [55]. The Pearson correlation coefficient is a widely used statistical measure for quantifying linear correlation between two variables. The linear relationship between variables $p$ and $q$ is defined as follows:

$$r = \frac{\text{Cov}(p, q)}{\sigma_p \sigma_q},$$

(9)

where $\text{Cov}(p, q)$ denotes the covariance and $\sigma$ the standard deviation.

To examine how the temporal dependence structure of the predicted inflows is preserved across different distance measures, the element-wise correlations are presented in Fig 11. The leftmost panel shows the element-wise correlation matrix of the original scenario set, while the others display the correlation matrices of the reduced sets obtained using the four distance measures. The correlation structure of the reduced set obtained using the energy distance exhibits the highest similarity to the original set, particularly evident in case 4 (Fig 11p to 11t). In contrast, the Wasserstein distance shows the most divergent correlation structure, and the Euclidean distance is similar to the Manhattan distance, as demonstrated in case 2 (Fig 11f to 11i).

For a detailed comparison, we calculate the mean absolute difference between the original scenarios' temporal correlation and the reduced sets' correlations for five different numbers of reduced scenarios ($m = 10, 20, 30, 40$, and 50). As depicted in Fig 12, the energy distance exhibits the smallest difference, while the Wasserstein distance demonstrates the largest deviation. As expected, the differences decrease as the number of reduced scenarios increases.

While Figs 11 and 12 provide visual comparisons, we further quantified the preservation of temporal dependencies by calculating the difference between the correlation matrices of the original and reduced sets. We employed the Frobenius norm of the difference between the correlation of distance measures ($||R_{original} - R_{reduced}||_F$) as a metric for structural divergence. This metric quantifies how well the model preserves the temporal correlations between times in the prediction horizon. Table 3 summarizes the averages of these errors. The Energy distance yielded the smallest error (0.69), indicating the highest fidelity in preserving the complex lag-dependence structure of the original scenarios. The Wasserstein distance showed the largest deviation, confirming the visual discrepancies observed in Fig 11. Furthermore, to explicitly evaluate lag-dependence preservation, we analyzed the Autocorrelation Function (ACF) in Fig 13. The results demonstrate that all measures seem to follow the same pattern, but with different values. The Energy distance (red) most closely tracks the original decay pattern, while the Manhattan (green) and Euclidean (blue) distances also adequately capture the correlations despite having slightly higher matrix errors.

From the computational point of view, the energy distance exhibits the highest complexity and lowest efficiency. This is because $n - i$ quadratic problems must be solved to select the $i^{th}$ scenario when applying a simple 1-step forward selection. As the number of reduced scenarios increases, the computational time increases linearly for the Wasserstein distance (from 11.8 seconds to 82.3 seconds), but scales quadratically for the energy distance (from 72.5 to 1787.3 seconds). Because of the exhaustive exploration of all potential combinations when incorporating the exact energy distance, the computational complexity would increase exponentially. The other distance measures are computationally efficient because of established efficient algorithms that do not need an explicit optimization process [16]. For the Manhattan and Euclidean distances, the computation time remains relatively constant regardless of increases in $m$, around 4 seconds for the Manhattan distance and 4.5 seconds for the Euclidean distance, as demonstrated in Table 4.

 

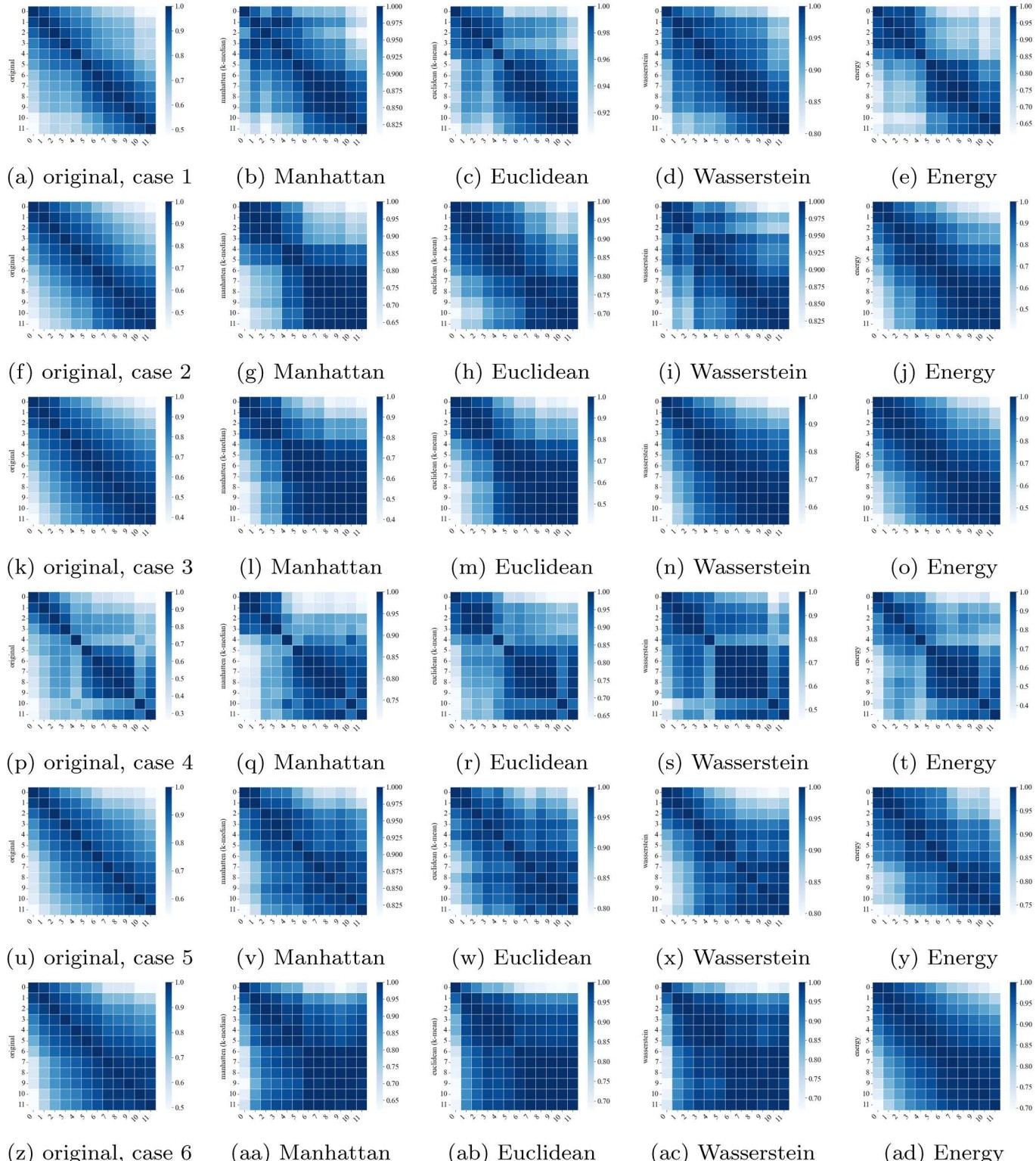

**Fig 11. Pearson correlation between inflows at different times when $m = 10$.**

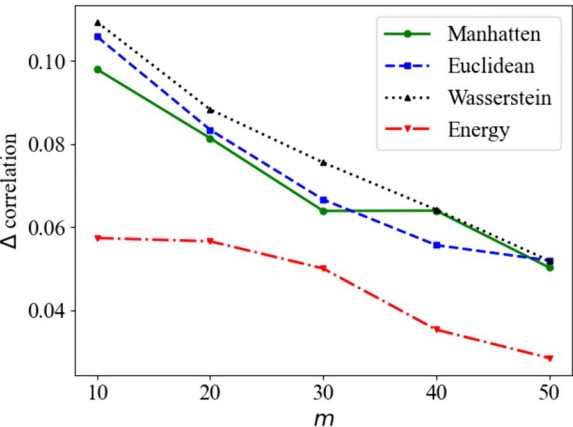

**Fig 12. Difference in correlations between original and reduced scenarios for different reduced scenario set sizes $m$, where the original set had 1000 scenarios.**

**Table 3. Frobenius norm of the difference distance measures.**

|  | Manhattan | Euclidean | Wasserstein | Energy |
|---|---|---|---|---|
| Frob. norm | 1.066 | 1.072 | 1.138 | 0.692 |

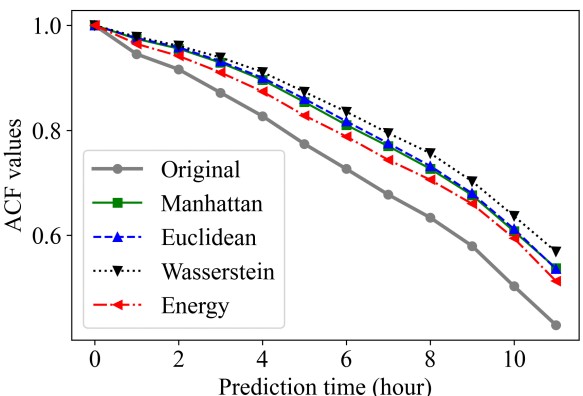

**Fig 13. Comparison of temporal autocorrelation functions (ACF) across different distance measures.** The energy distance (red) exhibits the highest fidelity to the original scenarios (gray), closely following its decay pattern. The Euclidean (blue) and Manhattan (green) distances also demonstrate robust performance, effectively capturing the temporal dynamics, whereas the Wasserstein distance (black) tends to overestimate temporal persistence.

## Discussion

This study established a probabilistic framework for reservoir flood control by applying Bayesian Neural Networks (BNNs) to generate inflow scenarios, verifying their applicability in capturing temporal dependencies in inflow time series. Here, the characteristics include both the temporal correlations and the accuracy degradation with increasing the prediction horizon. The MC dropout BNN model is found to effectively generate uncertain inflow scenarios, with reasonable performance demonstrated in terms of RMSE and NSE. Without explicitly defining temporal correlations between elements in a

**Table 4. Computational times for four distance measures (seconds).**

| $m$ | Manhattan | Euclidean | Wasserstein | Energy |
|-----|-----------|-----------|-------------|--------|
| 10 | 4.2 | 4.0 | 11.8 | 72.5 |
| 20 | 3.8 | 4.0 | 23.8 | 308.7 |
| 30 | 3.8 | 4.7 | 39.0 | 712.0 |
| 40 | 3.8 | 5.0 | 62.7 | 1131.5 |
| 50 | 4.0 | 5.0 | 82.3 | 1787.3 |

scenario, it reproduces the characteristic behavior of hydrological prediction, namely the degradation of performance with increasing prediction horizon (Fig 4).

However, our BNN model exhibited limitations in accurately predicting peak inflow (Fig 5). The limitations are primarily attributed to insufficient flood event data being used for training. The training dataset predominantly consists of 'normal' inflows characterized by small and short-term increases. Extreme values, such as peak flood inflow, are located at the tail of the data distribution, making accurate prediction difficult. When the model is trained predominantly on lower inflow values, it struggles to predict extreme values, leading to overfitting on small inflows and underfitting on extreme events [56,57]. Consequently, the model tends to underpredict the mean of extreme events. While this risk is mitigated in our framework by utilizing the probabilistic upper bounds, future studies could aim to improve the mean accuracy. Potential approaches include employing transfer learning [58] or regime-switching multi-model frameworks to explicitly capture high-flow dynamics [59]. However, such architectural enhancements are outside the scope of this study.

Additionally, the relatively low 95% PICP (Table 2) suggests the model is under-dispersed. This stems from employing the standard MSE loss, which implicitly assumes homoscedasticity and limits the capturing of data-dependent aleatoric uncertainty. While heteroscedastic loss functions (e.g., GNLL) could address this, we prioritized model robustness over complexity given the limited dataset. Therefore, adopting such loss functions remains a promising direction for future research to further refine uncertainty quantification.

Despite these limitations, our results demonstrate that the actual peak inflow is successfully captured under or near the upper envelope. This observation underscores the critical need to capture the full spread of uncertainty rather than relying solely on the mean. Consequently, this validates our specific evaluation criteria for scenario reduction: whether a given method can effectively preserve these extreme scenarios (the envelope) to ensure that the operational risk space is maintained (Fig 6).

The advantages and disadvantages of existing scenario reduction methodologies, particularly from the perspective of distance measures, are evaluated. The performances are assessed by three criteria: preservation of statistical properties, inclusion of extreme events, and computational complexity. Scenario reduction using the energy distance demonstrates superior performance in preserving the statistical properties of the original scenario set generated by the BNN model. The original temporal correlation structure is best maintained by the reduced scenarios based on the energy distance (Fig 12). Although all distance measures effectively preserve mean values, the energy distance achieves the minimum MAE for both mean and standard deviation metrics (Fig 10b and 10d).

Consideration and analysis of extreme scenarios are crucial for flood control. The span between the upper and lower envelopes of original scenarios represents the flood risk space resulting from uncertain inflows. To take into account this span, we suggest the size of the envelope as a metric for assessing the reduced scenarios using the $l_1$-norm of scenarios. For this, the Manhattan and Euclidean distances demonstrate superior performance (Fig 8 and 9). Moreover, in terms of temporal correlation and standard deviation, both the Manhattan and Euclidean distances exhibit comparable performance, following the energy distance (Fig 12 and 10d). No substantial differences are observed between the Manhattan and Euclidean distances in terms of both statistical properties and extreme scenario preservation.

However, from the computational efficiency perspective, the use of the energy distance demonstrates significant limitations compared to other distance measures (Table 4). Even with only 10 scenarios, the computation time exceeds one minute, extending to approximately 12 minutes for $m = 30$. Given that the time horizon is usually less than an hour for flood control, a scenario reduction process requiring more than 10 minutes proves impractical [1,60]. This is because the control inputs produced by an optimal control approach cannot be directly implemented in reservoir flood control due to their substantial impact on basin flood conditions [1]. Additionally, the process requires time for decision-making and information sharing with relevant organizations.

Consequently, considering the importance of extreme scenarios for optimal flood operation, the Manhattan and Euclidean distances can be reasonable and practical choices for scenario reduction. However, when preserving the statistical properties, such as mean, standard deviation, and temporal correlation, is prioritized over including extreme scenarios, and computational time is not a critical concern, the energy distance can be the best choice. Regardless of the distance measure selected, 30 reduced scenarios can be sufficient in terms of both inclusion of extreme events and preservation of statistical properties.

Although no prior work has directly evaluated scenario generation and reduction methodologies for reservoir flood control, a similar methodological framework has been applied in the domain of electricity demand and price forecasting [5]. In the study, energy-distance–based scenario reduction has been shown to preserve distributional properties more effectively than the widely used Wasserstein distance, albeit with higher computational cost due to its quadratic programming formulation. Comparable patterns are observed in our study, where the energy distance achieves the highest preservation of statistical properties in hydrological scenarios. However, for flood control, a non-statistical metric that measures the envelope of scenarios as a possible inflow volume metric has been introduced. Therefore, it is important to make a distinction in the performance of scenario reduction approaches based on the relevant metric tailored to the application (e.g., flood risk is related to total inflow volume). In our example, this reveals that the Manhattan and Euclidean distances outperform the other measures in retaining extreme inflow event properties, particularly for small $m$. Furthermore, by explicitly assessing computational feasibility under flood control time constraints, this study shows that energy distance may be impractical for fast real-time applications despite its statistical advantages—an operational consideration not addressed in [5].

It should be noted that we employ a simple 1-step forward selection method for the Wasserstein and energy distances. Even though it has been widely applied due to its computational efficiency, this algorithm cannot guarantee the selection of the optimal reduced set. To find the optimal reduced set closest to the original scenario set, all possible subsets should be explored. This is generally computationally intractable. However, the implementation of high-performance computing and/or parallelization techniques could potentially allow us to obtain the optimal reduced set based on the energy or Wasserstein distance with significantly reduced computational time. It remains to be determined whether this algorithm can improve the performance of these distance measures and whether reduced computational time is practical.

## Conclusions

This study demonstrated that Bayesian Neural Networks (BNNs) can be used to effectively generate probabilistic flood scenarios while capturing temporal dependencies in inflow time series. A comparative evaluation of four distance measures, i.e., the Manhattan, Euclidean, Wasserstein, and energy distance metrics, showed that the energy distance best preserves statistical properties, while the Manhattan and Euclidean distances are more effective in retaining extreme inflow events. Contrary to its widespread use in other applications, the Wasserstein distance offered no substantial advantage in this application where temporal dependencies of the time series are important.

From a practical perspective, the Manhattan and Euclidean distances provide a strong balance between preserving extreme events and maintaining computational efficiency, making them suitable for real-time reservoir flood control. The energy distance, although statistically superior, scales much less favourably computationally with the number of scenarios in the reduced set and is therefore less viable for time-sensitive operations.

The main limitation of this study is that the operational importance of retaining extreme scenarios was not tested through direct integration into reservoir flood control optimization, such as stochastic MPC. In addition, the analysis was based on a limited number of historical flood events, which may affect the generalizability of the findings. Future research should integrate the proposed scenario generation and reduction framework into real-time stochastic reservoir operation models to quantify operational benefits, explore hybrid approaches that combine the statistical strengths of the energy distance with the efficiency of the Manhattan or Euclidean distances, and assess performance using larger and more diverse flood event datasets to enhance robustness and transferability.

## Acknowledgments

The hydrological data were acquired from the public data portal (www.data.go.kr) and Korea Water Resources Public Corporation's website (http://kwater.or.kr). The data and code of this research are available at https://doi.org/10.4121/e343331b-496f-40ab-83eb-f546df6dffa6 under the CC-BY-4.0 licence.

## Author contributions

**Conceptualization:** Ja-Ho Koo, Edo Abraham, Andreja Jonoski, Dimitri P. Solomatine.

**Data curation:** Ja-Ho Koo.

**Formal analysis:** Ja-Ho Koo.

**Investigation:** Ja-Ho Koo.

**Methodology:** Ja-Ho Koo.

**Project administration:** Edo Abraham.

**Software:** Ja-Ho Koo.

**Supervision:** Edo Abraham, Andreja Jonoski, Dimitri P. Solomatine.

**Validation:** Ja-Ho Koo.

**Visualization:** Ja-Ho Koo.

**Writing – original draft:** Ja-Ho Koo.

**Writing – review & editing:** Edo Abraham, Andreja Jonoski, Dimitri P. Solomatine.

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
