## [Decision Letter · Decision Letter 0]

28 Apr 2025

PONE-D-25-16551Comparison of scenario reduction approaches for reservoir inflow timeseries generated by a Bayesian Neural NetworkPLOS ONE

Dear Dr. Koo,

Thank you for submitting your manuscript to PLOS ONE. After careful consideration, we feel that it has merit but does not fully meet PLOS ONE’s publication criteria as it currently stands. Therefore, we invite you to submit a revised version of the manuscript that addresses the points raised during the review process.

We look forward to receiving your revised manuscript.

Kind regards,

Namal Rathnayake, Ph.D.

Academic Editor

PLOS ONE

Journal Requirements:

2. We note that Figure 1 in your submission contain [map/satellite] images which may be copyrighted. All PLOS content is published under the Creative Commons Attribution License (CC BY 4.0), which means that the manuscript, images, and Supporting Information files will be freely available online, and any third party is permitted to access, download, copy, distribute, and use these materials in any way, even commercially, with proper attribution. For these reasons, we cannot publish previously copyrighted maps or satellite images created using proprietary data, such as Google software (Google Maps, Street View, and Earth). For more information, see our copyright guidelines: http://journals.plos.org/plosone/s/licenses-and-copyright.

Reviewers' comments:

Reviewer's Responses to Questions

**Comments to the Author**

1. Is the manuscript technically sound, and do the data support the conclusions?

Reviewer #1: Yes

Reviewer #2: Yes

2. Has the statistical analysis been performed appropriately and rigorously? 

Reviewer #1: Yes

Reviewer #2: Yes

3. Have the authors made all data underlying the findings in their manuscript fully available?

Reviewer #1: Yes

Reviewer #2: Yes

4. Is the manuscript presented in an intelligible fashion and written in standard English?

Reviewer #1: Yes

Reviewer #2: Yes

5. Review Comments to the Author

Reviewer #1: Rewrite the abstract properly by following structure: Background, Objective, Methods, Results, Conclusion.

Avoid lengthy descriptions like "we evaluate the applicability of the four distance measures widely used in the literature"

Rewrite the objectives separately and clearly

Focus more on gaps in hydrological scenario reduction.

Add a clear novelty statement like "To the best of our knowledge, ...

Clarify the hyperparameter optimization process. Don't just say TPE is used

No proper discussion on missing data handling or data cleaning steps

Merge figures or reduce redundancy

In Discussion,

There is no critical comparison with past scenario reduction studies

Discuss more about limitations of BNN for peak inflow predictions

Add clear recommendation for practitioners on when to use which distance measure.

Rewrite long sentences or maybe Split into two clear sentences.

Read these studies to further enrich your study. (DOI: 10.1016/j.asoc.2023.110722) Provides a soft computing approach for hydrological prediction which can complement BNN in uncertainty modeling. (DOI: 10.1007/s11269-025-04142-5) Demonstrates different ML models for water prediction, useful to compare or benchmark scenario generation models. (DOI: 10.2166/wpt.2024.147) Discusses ML-based hydrological modeling that can supplement your scenario generation techniques. (DOI: 10.5772/intechopen.1006491) Discusses novel fuzzy-based models that can be integrated or compared with BNN-based scenario reduction.

These studies are suggested for you to check the content and improve your paper. IF THEY ARE APPROPRIATE ONLY.

Reviewer #2: To further strengthen the manuscript, we suggest a few improvements. First, while the authors have described the data sources and provided processed datasets, a more detailed explanation of the data preprocessing steps—such as how missing values and anomalies were handled—would enhance the reproducibility and transparency of the study. Second, although the comparison of scenario reduction methods is comprehensive, the current setup associates each distance metric with a different reduction algorithm (e.g., clustering for Manhattan and Euclidean distances, forward selection for Wasserstein and energy distances). To isolate the pure effect of the distance metrics, it would be valuable to apply a uniform reduction algorithm across all metrics. Third, while the study clearly recommends practical distance measures considering computational constraints, it could further benefit practitioners by providing specific operational guidelines, such as recommended numbers of reduced scenarios under given time limits. Overall, these additions would make the already strong manuscript even more robust and practically impactful.

6. PLOS authors have the option to publish the peer review history of their article (what does this mean?). If published, this will include your full peer review and any attached files.

Reviewer #1: No

Reviewer #2: No

---

## [Author Response · Author response to Decision Letter 1]

5 Jun 2025

We sincerely appreciate thoughtful comments.

All concerns have been carefully considered and addressed in our revised manuscript.

The details of the revision can be found in the review response.

Thank you again for your effort and time.

Sincerely yours,

Ja-Ho Koo

---

## [Decision Letter · Decision Letter 1]

9 Jul 2025

PONE-D-25-16551R1Comparison of scenario reduction approaches for reservoir inflow timeseries generated by a Bayesian Neural NetworkPLOS ONE

Dear Dr. Koo,

Thank you for submitting your manuscript to PLOS ONE. After careful consideration, we feel that it has merit but does not fully meet PLOS ONE’s publication criteria as it currently stands. Therefore, we invite you to submit a revised version of the manuscript that addresses the points raised during the review process.

We look forward to receiving your revised manuscript.

Kind regards,

Babak Mohammadi

Academic Editor

PLOS ONE

Additional Editor Comments:

Three reviewers evaluated this manuscript. Please consider their comments carefully during the revision process. Additionally, take into account the following points:

1. Add a flowchart in the methodology section to illustrate the workflow of the current study.

2. Ensure that each table and figure is clearly explained and discussed in the main text. The values presented in the tables should be mentioned and discussed in the text. Figures also require more detailed discussion in the text.

**Comments from PLOS Editorial Office:** We note that one or more reviewers has recommended that you cite specific previously published works during the current round and in an earlier round of revision. As always, we recommend that you please review and evaluate the requested works to determine whether they are relevant and should be cited. It is not a requirement to cite these works and you may remove them before the manuscript proceeds to publication. We appreciate your attention to this request.

Reviewers' comments:

Reviewer's Responses to Questions

**Comments to the Author**

1. If the authors have adequately addressed your comments raised in a previous round of review and you feel that this manuscript is now acceptable for publication, you may indicate that here to bypass the “Comments to the Author” section, enter your conflict of interest statement in the “Confidential to Editor” section, and submit your "Accept" recommendation.

Reviewer #2: (No Response)

Reviewer #3: All comments have been addressed

Reviewer #4: (No Response)

2. Is the manuscript technically sound, and do the data support the conclusions?

Reviewer #2: Partly

Reviewer #3: Yes

Reviewer #4: Yes

3. Has the statistical analysis been performed appropriately and rigorously? 

Reviewer #2: I Don't Know

Reviewer #3: Yes

Reviewer #4: Yes

4. Have the authors made all data underlying the findings in their manuscript fully available?

Reviewer #2: Yes

Reviewer #3: Yes

Reviewer #4: Yes

5. Is the manuscript presented in an intelligible fashion and written in standard English?

Reviewer #2: Yes

Reviewer #3: Yes

Reviewer #4: Yes

6. Review Comments to the Author

Reviewer #2: Although the contribution is significant, the evidence supporting the claims is weak and the conclusions are currently unconvincing, so more substantial revisions are needed.

1. There have been no cases in which the generated and reduced scenarios have been applied to actual stochastic optimal control, and their actual effectiveness has not been fully proven.

For this reason, it is unclear whether the proposed method will directly lead to improved operational performance. Related papers must be cited to make claims.

2. The accuracy of peak prediction in real data is clearly low, raising doubts about the reliability of BNN in flood response. This is due not only to the limitations of the model but also to bias in the training data, so data selection and pretreatment must be reconsidered.

Reviewer #3: All changes are included in the manuscript...........................................................

Reviewer #4: This study focuses on generating reservoir inflow scenarios using a Bayesian Neural Network and evaluating methods for reducing these scenarios. This topic is important for practical water resources applications, particularly in reservoir flood control optimization. The revised manuscript is meaningful and contributes valuable insights to the field, with results that adequately support its intended contribution. The paper can be recommended for publication after the following revisions:

1. The Abstract is too long and includes excessive technical detail that would be more appropriate in the main text. There is redundancy, particularly around scenario generation and reduction concepts, which are repeated several times. Consider reducing redundancy and shortening it to under 300 words for clarity and conciseness.

2. The manuscript uses “NSC” as an abbreviation for Nash-Sutcliffe Efficiency, which is not standard. Please use the widely accepted abbreviation “NSE” for clarity and consistency with the literature. Consider citing some updated papers and incorporating the performance metrics discussed in these studies to strengthen the evaluation of model performance.

3. The spelling and capitalization of method names and their abbreviations should be carefully reviewed to ensure consistency and alignment with common literature standards. Abbreviations should be defined in full at their first occurrence, followed by the abbreviation in parentheses. Subsequent mentions should use only the abbreviation.

4. The manuscript, especially the Abstract, frequently uses the active voice (e.g., “we develop,” “we evaluate”). It is generally recommended to use the passive voice to improve objectivity and better align with conventional academic style.

5. The Introduction lacks a clear, concise statement of the study’s novelty and specific contribution within reservoir inflow scenario generation and reduction.

6. The literature review is too detailed and occasionally unfocused. Consider condensing the literature review by focusing on the most relevant studies and directly linking them to the research gap addressed in this work. Consider including recent studies on sustainable flood risk management to better connect the practical relevance of inflow uncertainty with broader flood management frameworks.

7. Add a flowchart in the Method section to visually summarize the proposed scenario generation and reduction steps for clearer understanding.

8. Some sentences throughout the manuscript are overly long or contain multiple ideas, which affect clarity and flow. It is recommended to simplify these sentences by breaking them into shorter, more focused statements.

9. Consider briefly comparing the findings with similar recent studies to better highlight the methodological contributions and contextual novelty of this work.

10. The Conclusion should briefly and impactfully summarize the key findings. The study’s limitations are not sufficiently addressed. More explicit methodological recommendations for future research should be provided.

7. PLOS authors have the option to publish the peer review history of their article (what does this mean?). If published, this will include your full peer review and any attached files.

Reviewer #2: No

Reviewer #3: No

Reviewer #4: No

---

## [Author Response · Author response to Decision Letter 2]

3 Sep 2025

Dear Editor and Reviewers,

We sincerely thank you for your valuable feedback and constructive suggestions, which have significantly improved the quality and readability of our manuscript. In the attached document "Response to Reviewers," we provide detailed responses to each comment and explain how we have addressed all suggestions in the revised manuscript.

The major revisions are summarized as follows:

1. A flowchart has been added to the Method section (Fig 1) to illustrate the research process more clearly.

2. The limitations of the study, including the exclusion of numerical experiments using stochastic optimization-based feedback control and the inaccuracy of BNN models in predicting peak inflows, have been presented and explained.

3. The abstract and conclusion have been carefully rewritten to improve clarity and readability by making them more concise and reducing redundancy.

Beyond addressing the specific reviewer comments, we have also made additional minor revisions throughout the manuscript to enhance clarity and readability while preserving the original meaning.

We hope that the revised manuscript now meets the journal's standards and look forward to your favorable consideration.

Sincerely yours,

Ja-Ho Koo

---

## [Decision Letter · Decision Letter 2]

19 Nov 2025

PONE-D-25-16551R2Comparison of scenario reduction approaches for reservoir inflow timeseries generated by a Bayesian Neural NetworkPLOS ONE

Dear Dr. Koo,

Thank you for submitting your manuscript to PLOS ONE. After careful consideration, we feel that it has merit but does not fully meet PLOS ONE’s publication criteria as it currently stands. Therefore, we invite you to submit a revised version of the manuscript that addresses the points raised during the review process.

We look forward to receiving your revised manuscript.

Kind regards,

Dr. Anurag Barthwal, Ph.D.

Academic Editor

PLOS ONE

**Journal Requirements:**

**Additional Editor Comments:**

Dear Authors,

Based on the Reviewer’s evaluations, the manuscript requires significant revisions before it can proceed further in the review process. Please address all reviewer comments carefully and submit a revised version of the manuscript along with a detailed point-by-point response explaining how each comment has been handled.

Kindly ensure that the revised manuscript highlights all changes clearly (using colored text) to facilitate further assessment.

(Academic Editor)

Reviewers' comments:

Reviewer's Responses to Questions

**Comments to the Author**

1. If the authors have adequately addressed your comments raised in a previous round of review and you feel that this manuscript is now acceptable for publication, you may indicate that here to bypass the “Comments to the Author” section, enter your conflict of interest statement in the “Confidential to Editor” section, and submit your "Accept" recommendation.

Reviewer #4: All comments have been addressed

Reviewer #5: All comments have been addressed

2. Is the manuscript technically sound, and do the data support the conclusions?

Reviewer #4: (No Response)

Reviewer #5: Partly

3. Has the statistical analysis been performed appropriately and rigorously? 

Reviewer #4: (No Response)

Reviewer #5: Yes

4. Have the authors made all data underlying the findings in their manuscript fully available?

Reviewer #4: (No Response)

Reviewer #5: Yes

5. Is the manuscript presented in an intelligible fashion and written in standard English?

Reviewer #4: Yes

Reviewer #5: Yes

6. Review Comments to the Author

**Reviewer #4:** Thank you for addressing all my previous comments clearly and convincingly. Some aspects could be further developed in future studies, but overall, the revised manuscript provides a valuable contribution to the field.

**Reviewer #5:** The work provides a systematic way to generate and reduce probabilistic inflow scenarios using machine learning and quantitative metrics, helping reservoir managers make better flood-control decisions under uncertainty.

The manuscript is technically rich and addresses a relevant hydrological challenge. However, I have following observations about the work:

1. The introduction contains extensive citations and detailed explanations of prior work, making it dense and difficult to follow. It can be streamlined without losing scientific depth.

2. Contribution 1 (BNN-based scenario generation) is presented as novel, but such approaches already exist in hydrology and ML; the novelty is more incremental than claimed. Pleas clarify.

3. There is no performance comparison with other deep learning models (such as LSTM, GRU, CNN), which weakens the justification for BNN. Authors are suggested to provide the performance comparison with other deep learning models.

4. The TPE algorithm is explained in textbook detail, which disrupts the methodological flow and adds complexity that could be summarized briefly.

5. The use of random validation sampling for time-series prediction is problematic. Temporal data requires chronological splitting to avoid information leakage.

6. Although k-means, k-median, and forward selection are used, the manuscript lacks a strong justification for using these specific algorithms over more modern or optimal alternatives.

7. Only 9 flood events exist in the dataset. This large imbalance between normal and extreme flows severely limits peak-performance learning, yet the issue is not addressed with corrective techniques (e.g., resampling, extreme-event augmentation).

8. The model underpredicts flood peaks—a major practical limitation for reservoir flood control—but no strategy to mitigate this is proposed.

9. Since the model is probabilistic, metrics such as CRPS, prediction interval coverage, or quantile errors should be included.

10. The envelope size used to evaluate extreme-event preservation is intuitive but not widely established in literature; more justification is needed.

11. Correlation matrices are compared visually, but deeper statistical interpretation (e.g., lag-dependence preservation) is missing.

7. PLOS authors have the option to publish the peer review history of their article (what does this mean?). If published, this will include your full peer review and any attached files.

Reviewer #4: No

Reviewer #5: **Yes:** Dr. Nikhil Kumar

---

## [Author Response · Author response to Decision Letter 3]

9 Dec 2025

We appreciate the valuable feedback provided by the reviewers. We have revised the manuscript accordingly. A detailed response to each comment can be found in the uploaded file labeled 'Review_response_JKoo_R3.pdf'.

---

## [Decision Letter · Decision Letter 3]

16 Mar 2026

PONE-D-25-16551R3Comparison of scenario reduction approaches for reservoir inflow timeseries generated by a Bayesian Neural NetworkPLOS One

Dear Dr. Koo,

Thank you for submitting your manuscript to PLOS ONE. After careful consideration, we feel that it has merit but does not fully meet PLOS ONE’s publication criteria as it currently stands. Therefore, we invite you to submit a revised version of the manuscript that addresses the points raised during the review process.

We look forward to receiving your revised manuscript.

Kind regards,

Ziqiang Zeng, Ph.D.

Academic Editor

PLOS One

Journal Requirements:

Reviewers' comments:

Reviewer's Responses to Questions

**Comments to the Author**

1. If the authors have adequately addressed your comments raised in a previous round of review and you feel that this manuscript is now acceptable for publication, you may indicate that here to bypass the “Comments to the Author” section, enter your conflict of interest statement in the “Confidential to Editor” section, and submit your "Accept" recommendation.

Reviewer #4: All comments have been addressed

Reviewer #6: (No Response)

2. Is the manuscript technically sound, and do the data support the conclusions?

Reviewer #4: (No Response)

Reviewer #6: Yes

3. Has the statistical analysis been performed appropriately and rigorously? 

Reviewer #4: (No Response)

Reviewer #6: Yes

4. Have the authors made all data underlying the findings in their manuscript fully available?

Reviewer #4: (No Response)

Reviewer #6: Yes

5. Is the manuscript presented in an intelligible fashion and written in standard English?

Reviewer #4: Yes

Reviewer #6: Yes

6. Review Comments to the Author

Reviewer #4: I thank the authors for addressing all previous comments. I have no further suggestions. I consider the manuscript suitable for publication in its current form.

Reviewer #6: The paper addresses a relevant and timely problem in stochastic reservoir flood control: how to select a compact set of inflow scenarios that preserves both statistical fidelity and extreme-event coverage while remaining computationally feasible for real-time operation. The methodology is well-grounded, the evaluation criteria are appropriate, and the results are thoroughly discussed. The following minor issues should be addressed before the paper is accepted for publication.

1) Line 5: "metrological uncertainties" should read "meteorological uncertainties." Metrology is the science of measurement, whereas meteorology is the science of weather and climate. Please correct this typo.

2) Lines 98-103: The section references in the closing paragraph of the Introduction are missing. The sentences "Section introduces the methodologies..." and "Section details the case study area..." lack their corresponding section numbers. Please correct.

3) Line 104: The acronym "MC" is used in the subsection heading before it is defined. It is only defined later, in line 124. Please define the acronym before its first use, or spell it out in the heading ("Monte Carlo dropout BNN").

4) Line 173: The definition of μᵢ in Eq 3 is inconsistent with the preceding text. Lines 160–162 correctly state that the k-median centroid is the element-wise median, whereas the k-means centroid is the element-wise mean. However, the caption of Eq 3 defines μᵢ as "the mean of all scenarios in the cluster Cᵢ" for both cases. For l = 1 (Manhattan distance, k-median), μᵢ should be the element-wise median, not the mean. Please correct the definition to distinguish the two cases, e.g., "the centroid of cluster Cᵢ (element-wise median for l = 1, element-wise mean for l = 2)."

5) Fig. 2: The URLs provided in the caption (www.data.or.kr and https://www.vworld.kr) could not be accessed. Please verify that these addresses are correct.

6) Lines 261-274: The training period begins in September 2012, while both test events precede this date (Event 1: August 2011, Event 2: August 2012). The rationale for starting training in September 2012 is not stated. Since data are reportedly available from 2011, a brief explanation of this choice would help readers understand whether this is a deliberate design decision (e.g., to ensure the test events are excluded from training) and its practical implications (i.e., the model is trained on data collected after the test events it is evaluated on).

7) Line 291: "when validation loss shows no improvement over a specified epochs" is grammatically incorrect. It should read "over a specified number of epochs."

8) Line 375: The size of the original scenario set is fixed at n = 1000. This choice is not justified. A brief explanation of the rationale—whether based on convergence of the empirical distribution, a pilot sensitivity study, or computational feasibility—would strengthen the methodology.

9) Line 498: A period is missing between "in the prediction horizon" and "Table 3." The two sentences run together without punctuation. Please insert a period after "prediction horizon."

10) Lines 510-511: The claim that "the computational time increases linearly as the number of reduced scenarios... increases" is inconsistent with both the Abstract and Table 4. The Abstract correctly states that the energy distance is "quadratic in m." Table 4 confirms this: when m increases fivefold (from 10 to 50), the energy distance computation time increases approximately 24.6-fold (from 72.5 to 1,787.3 s), which is consistent with quadratic scaling, not linear. Please correct line 511 to be consistent with the Abstract and the empirical data in Table 4.

11) Lines 521-646 (Discussion and Conclusions): Several minor language issues were found in these sections:

- Line 572: Figures should be cited in numerical order: "Figs 8 and 9" instead of "Fig 9 and 8."

- Line 606: "the Manhattan and Euclidean distances outperform in retaining" is missing a comparator. Please rephrase, e.g., "outperform the other measures in retaining."

- Line 613: "can not" should be written as one word: "cannot."

- Line 628: "other application" should read "other applications."

- Lines 634-635: "scales much less favourably computationally with the number of scenarios in the reduced set and as such less viable for time-sensitive operations" lacks a conjugated verb for the second predicate. Please rephrase, e.g., "... and is therefore less viable for time-sensitive operations."

7. PLOS authors have the option to publish the peer review history of their article (what does this mean?). If published, this will include your full peer review and any attached files.

Reviewer #4: No

Reviewer #6: No

---

## [Author Response · Author response to Decision Letter 4]

19 Mar 2026

We greatly appreciate the feedback and suggestions, which we believe have allowed us to improve the quality and readability of the manuscript. Our detailed responses are presented in the attached file, named 'Review response'.

---

## [Decision Letter · Decision Letter 4]

11 May 2026

Comparison of scenario reduction approaches for reservoir inflow timeseries generated by a Bayesian Neural Network

PONE-D-25-16551R4

Dear Dr. Ja-Ho Koo,

We’re pleased to inform you that your manuscript has been judged scientifically suitable for publication and will be formally accepted for publication once it meets all outstanding technical requirements.

Kind regards,

Nezir Aydin, Ph.D.

Academic Editor

PLOS One

---

## [Editor Report · Acceptance letter]

PONE-D-25-16551R4

PLOS One

Dear Dr. Koo,

I'm pleased to inform you that your manuscript has been deemed suitable for publication in PLOS One. Congratulations! Your manuscript is now being handed over to our production team.

Kind regards,

on behalf of

Professor Nezir Aydin

Academic Editor

PLOS One